# Vaccine Uptake to Prevent Meningitis and Encephalitis in Shanghai, China

**DOI:** 10.3390/vaccines10122054

**Published:** 2022-11-30

**Authors:** Hairenguli Maimaiti, Jia Lu, Xiang Guo, Lu Zhou, Linjie Hu, Yihan Lu

**Affiliations:** 1Department of Epidemiology, Ministry of Education Key Laboratory of Public Health Safety, School of Public Health, Fudan University, Shanghai 200032, China; 2Department of Immunization Planning, Minhang District Center for Disease Control and Prevention, Shanghai 201101, China; 3Institute of Immunization Planning, Shanghai Municipal Center for Disease Control and Prevention, Shanghai 200336, China

**Keywords:** meningitis, encephalitis, meningococcal vaccine, Japanese encephalitis vaccine, pneumococcal vaccine, *Haemophilus influenzae* type b (Hib) vaccine, rotavirus vaccine, enterovirus 71 vaccine, health belief model, structural equation model

## Abstract

Multiple vaccines may prevent meningitis and encephalitis (M/E). In China, the meningococcal vaccine and Japanese encephalitis vaccine (JEV) have been included in the expanded program of immunization (EPI). The pneumococcal vaccine, *Haemophilus influenzae* type b (Hib) vaccine, rotavirus vaccine, and enterovirus 71 (EV-71) vaccine are non-EPI vaccines and are self-paid. We aim to investigate the uptake of these M/E vaccines in children and the related knowledge and health beliefs among family caregivers. A total of 1011 family caregivers with children aged 1–6 years in Shanghai, China were included in the study. The uptake of the pneumococcal vaccine, Hib-containing vaccine, rotavirus vaccine, and EV-71 vaccine remained at 44.0–48.1% in children, compared with the higher uptake of the meningococcal vaccine (88.8%) and JEV (87.1%). Moreover, family caregivers had limited knowledge on the M/E pathogens and possible vaccines. Their health beliefs were moderate to high. Then, a health belief model (HBM) and a structural equation model were established. The uptake of four non-EPI vaccines was significantly influenced by family income (β = 0.159), knowledge (β = 0.354), self-efficacy (β = 0.584), and perceived susceptibility (β = 0.212) within an HBM. Therefore, it warrants further improving the uptake rate for these non-EPI vaccines to prevent potential M/E in children. A specific health promotion may empower the caregivers’ decision-making on childhood vaccination.

## 1. Introduction

Meningitis and encephalitis (M/E) cause a large disease burden worldwide. In 2019, the global incidence of meningitis was estimated to be 32.40 per 100,000 (95% confidence interval, 27.31–38.63), and that of encephalitis was 18.69 per 100,000 (16.54–20.87) [1,2]. Furthermore, M/E in children remains a critical public health concern, as it may cause significant mortality (3.9 per 100,000 for meningitis and 1.4 per 100,000 for encephalitis), fatality (up to 37%), and severe neurological sequelae (7.5–42%) such as physical and intellectual disability [3,4,5,6]. In China, the incidence of meningitis and encephalitis was reported to be 3.35 per 100,000 (2.74–3.98) and 15.27 per 100,000 (13.23–17.47), respectively [7]. However, a multicenter population-based surveillance study in China found that an M/E incidence was 30.8–96.9 per 100,000 among children < 5 years of age [8]. Another study in western China reported that the mortality of M/E was 3.1% [9], which was significantly higher in children with pneumococcal meningitis (18.9%) [10] or Japanese encephalitis (9%) [11]. Therefore, it suggests that M/E remains a public health priority.

The World Health Organization (WHO) recommends the meningococcal vaccine and Japanese encephalitis vaccine (JEV) in the expanded program of immunization (EPI) to prevent M/E, especially in countries with a high disease burden. A meningococcus vaccination has been implemented in the national immunization programs of 43 countries [12]. However, the vaccination strategies differ among countries, due to diverse groups of meningococcus and disease burden, cost–benefits, budget impact, and public health priority [13,14]. Two categories of meningococcal vaccines are currently available in China: (1) meningococcal polysaccharide vaccines (MPSV), such as serogroup A MPSV (MPSV-A) and serogroup A and C (MPSV-AC) that are EPI vaccines, and serogroup A, C, Y, and W135 (MPSV-ACYW135) that is non-EPI vaccine, and (2) the meningococcal polysaccharide conjugate vaccine (MPCV) that is a non-EPI vaccine, including MPCV-AC and MPCV-ACYW135. In addition, 15 of 24 (62%) countries with JEV transmission risk have included a JEV vaccination in the national or regional programs of immunization [6]. In China, live attenuated JEV is implemented in 29 provinces as an EPI vaccine (except three western provinces as non-endemic areas), while inactivated JEV is optional for children as non-EPI. Generally, the uptake of MPSV and JEV in EPI are more than 90% in China, while that of non-EPI MPSV/MPCV and JEV differs across the regions [15].

Additionally, more than 100 different pathogens have been recognized as causative agents of M/E [16]. *Streptococcus pneumonia*, *Haemophilus influenzae* type b (Hib), rotavirus, and enterovirus 71 (EV-71) may invade the central nervous system (CNS) to cause M/E in children, in addition to infection in the respiratory tract and gastrointestinal tract. Accordingly, vaccines for these pathogens may moderately reduce the incidence of M/E, such as pneumococcal meningitis in Togo [17], Burkina Faso [18] and Spain [19], Hib meningitis in Morocco [20], rotavirus-associated encephalitis and encephalopathy in Japan [21], and EV-71-associated neurological complications in China [22]. Thus, a vaccination against M/E remains crucial for Chinese children. However, all vaccines for those four pathogens are non-EPI, demonstrating a moderate to low uptake rate in China [23,24,25,26,27]. In this study, in Shanghai, a highly developed metropolis in China, we aim to investigate the uptake of the above vaccines that may prevent M/E in children, the knowledge and health beliefs of M/E, M/E vaccines, and vaccination among family caregivers, which would facilitate improving vaccination strategies for those vaccines.

## 2. Materials and Methods

### 2.1. Study Design

This study focused on six vaccines that may prevent M/E (M/E vaccines): (1) MPSV/MPCV; (2) JEV; (3) 13-valent pneumococcal conjugate vaccine (PCV13)/23-valent pneumococcal polysaccharide vaccine (PPSV23); (4) Hib-containing vaccine, including Hib vaccine, diphtheria-tetanus-pertussis (DTP)-Hib vaccine, and DTP-inactivated polio vaccine (IPV)-Hib vaccine; (5) rotavirus vaccine; and (6) EV-71 vaccine (Table 1). Six M/E vaccines that were available in China’s mainland and their manufacturers were listed in Appendix A. In China, EPI vaccines are free and mandatory to children, while non-EPI vaccines are self-paid and voluntary. Furthermore, non-EPI vaccines have two categories: surrogate vaccines, such as MPCV and inactivated JEV that may replace MPSV and live attenuated JEV, respectively, and non-surrogate vaccines, such as the Hib-containing vaccine, rotavirus vaccine, and EV-71 vaccine, which have not been included in the EPI regardless of vaccine type.

We conducted a survey among the family caregivers using an online questionnaire during January and April 2022 in Shanghai, China. The inclusion criteria were as follows: (1) family caregivers had children aged 1–6 years and (2) their children had completed or were eligible for the immunization schedules of six M/E vaccines. Children who had contradictions for these vaccines were excluded in the study.

In the six M/E vaccines, MPSV/MPCV and JEV include both EPI and non-EPI vaccines, whereas the PCV13/PPSV23, Hib-containing vaccine, rotavirus vaccine, and EV-71 vaccine are non-EPI which has a lower uptake rate [24,26]. Then, we determined the sample size by using the uptake rate of the four non-surrogate non-EPI vaccines, with the procedure test for one proportion in the PASS 15.0 (NCSS, LLC, Kaysville, UT, USA) [28]. We set up P0 as 0.4 as described elsewhere [29], and assumed P1 to be 0.45, with power = 0.90 and α = 0.05. The sample size was estimated to be 1022.

### 2.2. Questionnaire Design and Distribution

The questionnaire investigated the following content: (1) demographics, such as the family caregivers’ relations to children (mother, father, or grandparents), age, gender, ethnicity, educational level, monthly family income, the number of children, and the children’s age; (2) history of communicable diseases in children, including M/E, pneumonia, diarrhea, and hand-foot-and-mouth disease (HFMD); (3) vaccination history of MPSV/MPCV, JEV, PCV13/PPSV23, Hib-containing vaccine, rotavirus vaccine, and EV-71 vaccine; (4) knowledge of clinical manifestations and pathogens of M/E, transmission routes of the pathogens, and diseases that may be prevented by the above vaccines; and (5) health beliefs towards M/E, M/E vaccines, and vaccination. An English version of the questionnaire is available in Appendix A.

A quick code that accessed the online questionnaire (supported by www.wjx.com (accessed on 26 November 2022)) was distributed to the family caregivers who attended the vaccination clinics in community health centers. In the questionnaire, respondents who intended to join the survey first read the informed consent, and then clicked “I agree to join the survey” to fill in the questionnaire. We selected 20 vaccination clinics in urban and suburban areas in Shanghai for the recruitment of respondents. In each vaccination clinic, we did not limit the respondents by sex, age, or any other demographics, which may ensure recruiting more potential respondents. Using the convenient sampling strategy, we finally recruited a total of 1175 respondents in the study.

### 2.3. Measurement of Outcomes

In this study, the respondents’ demographics and history of communicable in children were measured as the categorical variables. Moreover, vaccination history was measured as a dichotomous variable. The age to initiate vaccination was less than 1 year for all six M/E vaccines (Table 1), which could ensure that children of the respondents should have completed the basic immunization schedule of these vaccines. Furthermore, MPSV/MPCV and JEV had both EPI (free) and surrogate non-EPI (self-paid) vaccines (Table 1). Thus, we classified the vaccination history of MPSV/MPCV and JEV to “vaccinated” and “not vaccinated”, regardless of EPI or non-EPI vaccines. For PCV13/PPSV23, the Hib-containing vaccine, rotavirus vaccine, and EV-71 vaccine that are non-EPI, we similarly classified the vaccination history to “vaccinated” and “not vaccinated”. The “not vaccinated” group included children who did not receive those vaccines but remained eligible in age, and those who were over the vaccination age.

In the knowledge section, we utilized multiple-choice questions to determine the knowledge. A total of ten pathogens were presented to ask the respondents whether these pathogens may cause M/E, including meningococcus, Japanese encephalitis virus, *Streptococcus pneumonia*, Hib, rotavirus, EV-71, herpes simplex virus, varicella-zoster virus, measles virus, and mumps virus (vaccines to the latter three pathogens are EPI vaccines with an uptake rate up to 90% and were not included in this study). One point was given for checking each pathogen. For the common clinical manifestations of M/E (1 question), the transmission routes of the above pathogens (5 choices for each pathogen: airborne, food-borne, close contact, mosquito-borne, and blood-borne) and diseases that may be prevented by the above vaccines (4 choices for each of the six vaccines: M/E, pneumonia, diarrhea, and HFMD); 3 points were given for checking all correct answers, 2 points for checking part correct answers, 1 point for both correct and incorrect answers, and 0 points for all incorrect answers. Then, the scores from the above questions were added together to get to total score, which ranges between 0 and 61.

Moreover, we constructed a health belief model (HBM) [30]. A total of 16 questions were prepared to determine the six constructs, including the perceived susceptibility, perceived severity, perceived benefits, self-efficacy (perceived confidence to conduct a certain behavior) [31], perceived barriers, and cues to action. A 5-point Likert scale was employed to score the answers, including strongly agree (5 points), agree (4 points), neutral (3 points), disagree (2 points), and strongly disagree (1 point). Of the above questions, C10, C11, C12, C13, and C14 were the negative attitude questions, for which 6 points were subtracted from the scores; the other positive attitude questions were directly scored so that all the questions had scores in the same direction. The total score of the HBM could range from 16 to 80.

### 2.4. Statistical Analysis

Descriptive statistics were presented to determine the demographics of the respondents, the history of communicable disease, and the uptake of M/E vaccines in children. A chi-square test, one-way ANOVA, and non-parametric test were utilized when applicable. Logistic regression was further employed to determine the association of vaccine uptake with scores of knowledge and health beliefs and demographics. The number of vaccine uptake was used as a weight to plot a chord diagram to show the interrelationship of a combination between the six vaccines using the chord diagram package of R 4.2.0 (R Core Team (2022). R: A language and environment for statistical computing. R Foundation for Statistical Computing, Vienna, Austria. URL https://www.R-project.org/(accessed on 26 November 2022)).

For the HBM, multiple measures of the reliability and validity of the items within the constructs were assessed. The reliability of the items within a construct was assessed using Cronbach’s α and composite reliability [32]. The construct validity was further examined using a KMO test and Bartlett’s test of sphericity [33]. The convergent validity was examined using the average variance extracted (AVE) factor loadings [34,35]. The discriminant validity was also assessed to check whether a construct was truly distinct from other constructs by comparing the cross loadings within and between the constructs [34].

A structural equation model (SEM) was established to determine the association among the variables, with an uptake of four non-surrogate non-EPI M/E vaccines as the internal variables and dimensions of related knowledge, health beliefs, and demographics as external variables. Collinearity between the independent variables was analyzed using the variance inflation factor (VIF). Then, the maximum likelihood method was utilized to establish the SEM. The final model was selected based on the fit statistics, including the root mean square error of the approximation (RMSEA), the goodness of fit index (GFI), the adjusted goodness of fit index (AGFI), the incremental fit index (IFI), the comparative fit index (CFI), the Tucker–Lewis index (TLI) and parsimony normed fit index (PNFI) [36]. Additionally, a path diagram of vaccine uptake was plotted using the SEM. For the path arrows, the digits were standardized regression coefficients and correlation coefficients for double arrows. The circles indicate the latent variables and the boxes represent the observed indexes.

All statistical analyses were performed using SAS 9.4 (SAS Institute Inc., Cary, NC, USA) and IBM SPSS AMOS 26.0 (IBM, Armonk, NY, USA). A *p* value < 0.05 was considered to be statistically significant.

### 2.5. Ethical Approval

This study was approved by the Institutional Review Board of the Fudan University School of Public Health (IRB 00002408 and FWA 00002399) under IRB #2021-05-0902.

## 3. Results

### 3.1. Vaccine Uptake

A total of 1175 family caregivers completed the questionnaire, of which 1011 (86.0%) had children aged 1-6 years. They were included for further analysis in the study. Among the respondents, 68.5% were mothers, 29.1% were fathers, and 2.4% were grandparents. The majority of them were Han Chinese (88.0%), had single children (50.7%), had an educational level of college and below (54.5%), and had a medium monthly family income (CNY 5000–9999) (34.3%). Furthermore, 388 (38.38%) reported a history of communicable diseases in their children. A history of pneumonia (*n* = 216; 21.4%) and diarrhea (*n* = 216; 21.4%) were more common in children, compared with less HFMD (*n* = 126; 12.5%) and M/E (*n* = 9; 0.9%) (*p* < 0.001).

The uptake of six M/E vaccines in children significantly differed among the respondents (P<0.001). The majority (90.1%) of children had received MPSV/MPCV (totally 88.8%, including 40.9% received free vaccines, 32.7% received self-paid vaccines, and 15.2% remained unclear of the types) and/or JEV (totaling 87.1%, including 45.2% who received free vaccines, 20.6% who received self-paid vaccines, and 21.4% who remained unclear of the types) (Figure 1). However, the uptake of four non-surrogate non-EPI vaccines was much lower. It was 48.1% for PCV13/PPSV23, 44.5% for the Hib-containing vaccine, 46.2% for the rotavirus vaccine, and 44.0% for the EV-71 vaccine.

Moreover, 28.4% of the respondents’ children had received a total of six M/E vaccines, 32.2% had received 3–5 vaccines, 30.8% had received two or fewer vaccines, while 8.7% reported an unclear history of vaccination for certain or of all the vaccines or of those they did not receive. The combination of MPSV/MPCV and the JEV uptake was the highest (85.9%) (Figure 2). The Kappa value for the consistency between choosing free/self-paid JEV and free MPSV/self-paid MPCV was 0.6028 (*p* < 0.0001). Furthermore, children who received self-paid MPCV/JEV were more likely to take at least one non-surrogate non-EPI vaccine, compared with those who chose free MPSV/JEV (*p* < 0.0001 for MPCV and *p* = 0.0060 for JEV). For the non-surrogate non-EPI vaccines, the combination of PCV13/PPSV23 and Hib-containing vaccine was the highest (37.1%), followed by PCV13/PPSV23 and the rotavirus vaccine (36.8%), and the rotavirus vaccine and EV-71 vaccine (36.0%).

### 3.2. Knowledge and Health Beliefs

We investigated the knowledge of M/E pathogens and vaccines. Among the respondents, 72.6% correctly recognized the clinical manifestations of M/E, whereas 22.1% partially recognized and 5.3% responded with “I don’t know”. In the M/E pathogens, meningococcus was the most widely recognized (93.0%), followed by Japanese encephalitis virus (59.9%), *Streptococcus pneumonia* (45.0%), and Hib (40.0%); in contrast, rotavirus (14.3%) and EV-71 (14.1%) were less recognized (Table 2). Compared to knowing these pathogens, the respondents were less likely to recognize the transmission routes of the pathogens (Table 2); for each pathogen, the difference was significant (each *p* < 0.001). Furthermore, the majority of respondents understood that M/E may be prevented by MPSV/MPCV (91.0%), JEV (85.2%), and Hib-containing vaccine (54.7%); however, fewer knew that M/E may be prevented by the rotavirus vaccine (34.4%), EV-71 vaccine (31.2%), and PCV13/PPSV23 (29.7%) (Table 2).

We further determined the six HBM constructs, including the perceived susceptibility, perceived severity, perceived benefits, self-efficacy, perceived barriers, and cues to action, using 16 subscale ratings. The proportion of “strongly agree” and “agree” were much lower in the responses to “C2 My child is very susceptible to getting M/E” (21.2%) (within the perceived susceptibility), the “C11 M/E vaccination service is inconvenient” (28.9%) and “C12 M/E vaccines may have long-term adverse effects in children” (24.5%) (within perceived barriers), whereas that was all above 50% in the other 13 responses (Figure 3).

Overall, the mean score of knowledge was determined to be 27.22 (standard deviation, SD, 6.22) and the mean score of the health beliefs was 56.33 (SD, 6.43). The scores in both knowledge and health beliefs were positively associated with a higher family income, higher educational level, more history of communicable diseases in children, a higher uptake of MPSV/MPCV and four non-surrogate non-EPI vaccines, and was negatively associated with children’s age (each *p* < 0.05) (Table 3). Additionally, knowledge was significantly higher in the respondents of a senior age (*p* = 0.0178), while the health beliefs were significantly higher in those who were Han Chinese (*p* = 0.0063), had a single child (*p* = 0.0111), and chose JEV (*p* < 0.0001) (Table 3).

### 3.3. Factors Associated with Vaccine Uptake

The uptakes of six M/E vaccines were determined to be positively associated with a higher family income, knowledge score (except JEV), and health belief score (each *p* < 0.05) (Table 4). In addition, a single child, less than 36 years of age, and being Han Chinese were associated with a higher uptake of certain vaccines.

Moreover, the validity and reliability were measured in the forms of Cronbach’s α, composite reliability, construct validity, convergent validity, and discriminant validity (Appendix A).

Then, we established an SEM to determine the effect of knowledge, HBM constructs, and demographics on the uptake of four non-surrogate non-EPI vaccines. In the initial screening, only monthly family income was retained in the model based on the fitness indices. The final model was identified given the good fit indices for all the samples (Table 5). It indicated that the knowledge score (β = 0.354), family income (β = 0.159), and two HBM constructs, self-efficacy (β = 0.584) and perceived susceptibility (β = 0.212), positively influenced the uptake of four non-surrogate non-EPI vaccines (Table 6, Figure 4). However, the other HBM constructs had no effect.

## 4. Discussion

This study identified that the uptake of the pneumococcal vaccine (PCV13/PPSV23), Hib-containing vaccine, rotavirus vaccine, and EV-71 vaccine remained at 44–48% in children, compared with the higher uptake of the meningococcal vaccine (MPSV/MPCV, 88%) and JEV (87%). It may be explained that meningococcal vaccines and JEV had both an EPI vaccine and surrogate non-EPI vaccine, while the four other vaccines are a non-surrogate non-EPI vaccine. Notably, we found that 28% of the respondents had their children vaccinated with all six M/E vaccines. It has been documented that 4.7–48.9% of M/E with a confirmed etiology are attributable to *Streptococcus pneumonia*, Hib, rotavirus, and EV-71 [4,38,39]. Additionally, the four non-surrogate non-EPI vaccines could effectively prevent pneumonia, diarrheal, and HFMD that have a high incidence in China. Therefore, it warrants further improving the uptake rate for these vaccines.

In previous studies, the uptake of PCV13/PPSV23, the Hib-containing vaccine, the rotavirus vaccine, and the EV-71 vaccine differed widely in diverse regions of China [24,26,40]. It may be attributable to multiple influencing factors, including vaccine safety and effectiveness, vaccine price, demographics of caregivers, knowledge and awareness, health conditions, and the medical history of children [26,41,42,43,44,45]. In our study, similar factors were determined, such as demographics, knowledge, and health beliefs; of them, family income was significantly associated with the uptake of all six M/E vaccines and had the largest impact (Table 4). Moreover, children who received free and self-paid JEV were more likely to receive free MPSV or self-paid MPCV, respectively, with a moderate consistency (Kappa value = 0.6028). Similarly, children who received self-paid MPCV and JEV were more likely to take at least one non-surrogate non-EPI vaccine (*p* < 0.0001 for MPCV and *p* = 0.0060 for JEV). The findings indicated that economic factors remain crucial for decision-making on childhood vaccination among family caregivers.

Moreover, vaccine-preventable disease and vaccine-related knowledge may contribute to vaccine uptake, such as in an influenza and COVID-19 vaccination [46,47,48]. We obtained consistent findings in our study, that knowledge score positively influenced the uptake of four non-surrogate non-EPI vaccines (Table 6, Figure 4). However, we also found that family caregivers had limited knowledge on the M/E pathogens and possible vaccines, such as 80% of the respondents not correctly knowing the transmission routes of meningococcus, 40% not knowing that the Japanese encephalitis virus causes M/E, and >60% not knowing that the pneumococcal vaccine, rotavirus vaccine, and EV-71 vaccine may prevent M/E. Furthermore, discordance was observed between the knowledge on the vaccines and pathogens, such as the amount of respondents which knew that the Japanese encephalitis vaccine may prevent M/E (85.2%) was more than those who knew about the Japanese encephalitis virus (59.9%). In routine health education, family caregivers may receive more information on the effect of vaccines (“what diseases may be prevented by vaccines”) than pathogens (“what pathogens may cause diseases”). In addition, the meningococcal vaccine and Japanese encephalitis vaccine have been known as an “epidemic M/E vaccine” and “category II M/E vaccine” in Chinese, where there is a Chinese abbreviation for M/E. It provides more information on the vaccine-preventable diseases than pathogens. We also identified similar knowledge scores between the EV-71 vaccine and rotavirus vaccine. Although the EV-71 vaccine substantially prevents the M/E, it shared similar knowledge to the rotavirus vaccine, suggesting the low knowledge of the EV-71 vaccine.

We determined the health beliefs using an HBM that has been performed in multiple vaccine studies, such as the HPV vaccine, pneumococcal vaccine, and COVID-19 vaccine [30,33,49,50]. In these studies, health beliefs, such as the perceived benefit and perceived barriers, were determined to be significantly associated with the refusal of the HPV vaccine, a lower uptake of the pneumococcal vaccine compared with the meningitis vaccine, or a willingness to receive a COVID-19 vaccine. In our study, the knowledge and health beliefs were both associated with the uptake of almost all vaccines. Self-efficacy could be explained as a perceived confidence to conduct a certain behavior (i.e., vaccination or other preventive measures) or the evaluation and judgment of the ability to solve the problems resulting from certain behavior (i.e., vaccination or other preventive measures) [31,51]. It had a positive association with the uptake of four non-surrogate non-EPI vaccines, suggesting that when respondents had a greater confidence on the impact of vaccination, they were more willing to receive the vaccines for their children. The perceived susceptibility also influenced the vaccine uptake, which corresponded to the caregivers’ limited knowledge on M/E pathogens and vaccines (even for meningococcus and Japanese encephalitis virus). It is noteworthy that the responses to the HBM questions were scored using a 5-point Likert scale, which demonstrated the attitudes towards health beliefs instead of classifying correct or incorrect answers. Our findings warrant improvement in health education and promotion among caregivers.

Additionally, certain combinations of the uptake of the M/E vaccine were observed in our study. For the non-surrogate non-EPI vaccines, the most common combination was PCV13/PPSV23 and the Hib-containing vaccine, followed by PCV13/PPSV23 and the rotavirus vaccine, and the rotavirus vaccine and EV-71 vaccine. It may be associated with demographics as well as vaccination age in the schedules. In China, non-EPI vaccines have been increased, such as MPCV-ACYW135, PCV13, the pentavalent rotavirus vaccine, and the EV-71 vaccine. For these vaccines, ages to initiate vaccination are overlapped, which might lead to difficulties in receiving more non-EPI vaccines at the same months of age. Further study may be performed on how to implement the coadministration strategies of the EPI vaccine and non-EPI vaccine or two non-EPI vaccines.

This study had strengths and limitations. First, we included four M/E vaccines, in addition to the meningococcal vaccine and JEV, for the analysis. It broadly illustrated the scenario of an M/E vaccination in children and family caregivers’ perception towards an M/E vaccination. Second, we identified the association among knowledge, health beliefs, and vaccine uptake. The findings would facilitate precisely providing evidence for improving the vaccination strategies of M/E vaccines, such as including the prevention of M/E in health education and the promotion for certain vaccines. However, this study was performed in Shanghai city with a limited sample size and generalizability, during the implementation of COVID-19 containment measures between January and April 2022, personal mobility was restricted, and the health service might be disrupted at a certain degree, which may lead to drops in childhood vaccination. The vaccination history was obtained only from respondents’ reports, which may result in recall bias. Additionally, as the pneumococcal vaccine, Hib vaccine, rotavirus vaccine and EV-71 vaccine are not M/E-specific, most of family caregivers had a very limited knowledge towards these vaccines.

## 5. Conclusions

The uptake of the PCV13/PPSV23, Hib-containing vaccine, rotavirus vaccine, and EV-71 vaccine remained at 44–48% in children in Shanghai. Family caregivers had a limited knowledge on the M/E pathogens and possible vaccines. In contrast, their health beliefs were moderate to high. Moreover, the uptake of these four vaccines was significantly influenced by family income, knowledge, and self-efficacy and perceived susceptibility within the HBM. It warrants a tailored improvement for each vaccination strategy to enhance the prevention of M/E in children.

## Figures and Tables

**Figure 1 vaccines-10-02054-f001:**
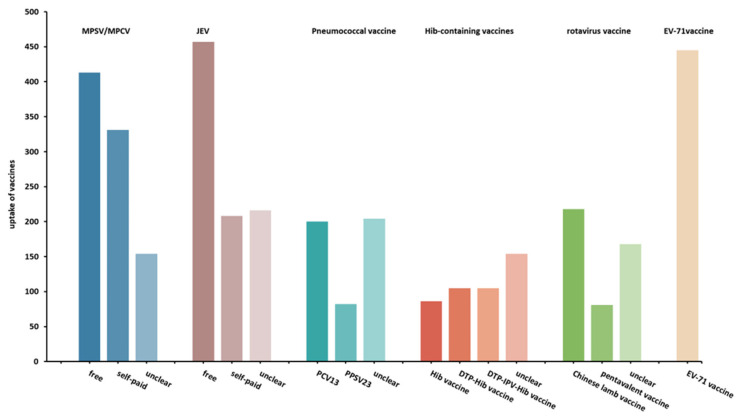
Uptake of six vaccines that may prevent meningitis/encephalitis in children. MPSV, meningococcal polysaccharide vaccine; MPCV, meningococcal polysaccharide conjugate vaccine; JEV, Japanese encephalitis vaccine; PCV13, 13-valent pneumococcal conjugate vaccine; PPSV23, 23-valent pneumococcal polysaccharide vaccine; Hib, *Haemophilus influenzae* type b; DTP, diphtheria-tetanus-pertussis vaccine; IPV, inactivated polio vaccine; EV-71, enterovirus 71.

**Figure 2 vaccines-10-02054-f002:**
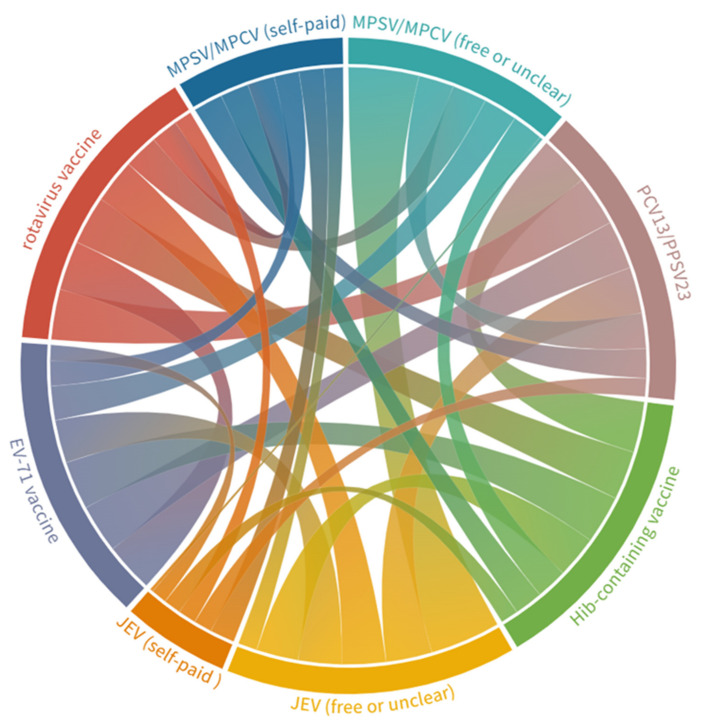
A chord diagram presenting the concurrent uptake of vaccines that may prevent meningitis/encephalitis in children. MPSV, meningococcal polysaccharide vaccine; MPCV, meningococcal polysaccharide conjugate vaccine; JEV, Japanese encephalitis vaccine; PCV13, 13-valent pneumococcal conjugate vaccine; PPSV23, 23-valent pneumococcal polysaccharide vaccine; Hib, *Haemophilus influenzae* type b; EV-71, enterovirus 71.

**Figure 3 vaccines-10-02054-f003:**
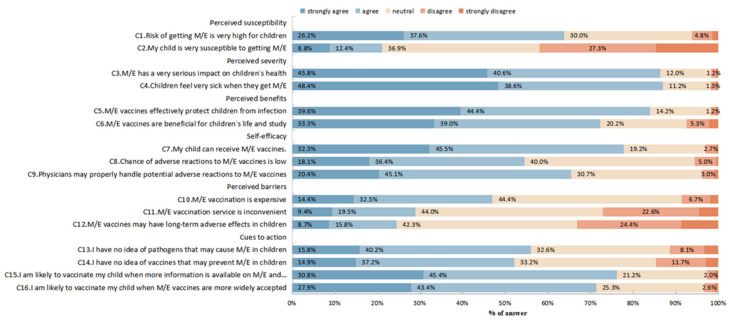
Responses to the health belief model (HBM) constructs.

**Figure 4 vaccines-10-02054-f004:**
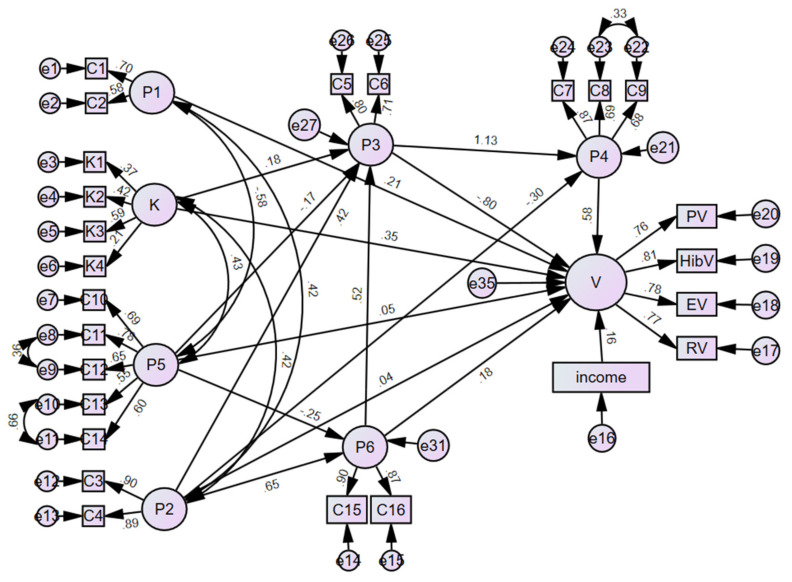
A path diagram of vaccine uptake using a structural equation model (sample size = 1011; model fitness in Table 5). For path arrows, the digits were standardized regression coefficients and correlation coefficients for double arrows. The circles indicated latent variables and boxes represented observed indexes. Income, monthly family income; K, knowledge; P1, perceived susceptibility; P2, perceived severity; P3, perceived benefits; P4, self_efficacy; P5, perceived barriers; P6, cues to action; V, vaccination uptake, PV, pneumococcal vaccine; HibV, *Haemophilus influenza*- containing vaccine; RV, rotavirus vaccine; EV, enterovirus 71 vaccine.

**Table 1 vaccines-10-02054-t001:** Immunization schedules of six vaccines that may prevent meningitis/encephalitis in children.

Vaccines	Types	Immunization Schedules
Meningococcal polysaccharide vaccine (MPSV)/meningococcal polysaccharide conjugate vaccine (MPCV)	EPI and non-EPI (surrogate)	Four doses administered at 6 months, 9 months, 3 years, and 6 years. EPI (free), non-EPI (self-paid) vaccines, or combined, are optional.Three doses of non-EPI vaccines at 6 months, 9 months, and 3 years are optional.
Japanese encephalitis vaccine (JEV) (live attenuated and inactivated vaccines)	EPI and non-EPI (surrogate)	Two doses, administered at 2 months and 2 years. EPI (live attenuated) or non-EPI (inactivated) vaccines are optional. Additionally, the third dose of the non-EPI vaccine is optional at 6 years.
13-valent pneumococcal conjugate vaccine (PCV13)/23-valent pneumococcal polysaccharide vaccine (PPSV23)	Non-EPI (non-surrogate)	PCV13: four doses administered at 1, 3, 5 months, and 1 year. PPSV23: one dose administered at 2 years.
*Haemophilus influenzae* type b (Hib)-containing vaccines (Hib vaccine, DTP-Hib vaccine, and DTP-IPV-Hib vaccine)	Non-EPI (non-surrogate)	Hib vaccine and DTP-IPV-Hib vaccine: four doses administered at 2, 3, and 4 months, and 1.5 years. DTP-Hib vaccine: four doses administered at 3, 4, and 5 months, and 1.5 years.
Oral live attenuated rotavirus vaccine (Chinese lamb rotavirus vaccine and pentavalent rotavirus vaccine)	Non-EPI (non-surrogate)	Chinese lamb vaccine: one dose administered for children aged from 2 months to 3 years, per year. Pentavalent vaccine: three doses administered at 1, 2, and 3 months.
Inactivated enterovirus 71 (EV-71) vaccine	Non-EPI (non-surrogate)	Two doses administered at 6 months and 7 months.

EPI, expanded program of immunization;
DTP, diphtheria-tetanus-pertussis vaccine; IPV, inactivated polio vaccine.

**Table 2 vaccines-10-02054-t002:** Knowledge of meningitis/encephalitis (M/E) and related vaccines.

Pathogens	Correct Answer (%)		
	Pathogens May Cause M/E	Possible Transmission Routes of Pathogens *	M/E May Be Prevented by Possible Vaccines **
Meningococcus	93.0	19.3	91.0
*Streptococcus pneumonia*	45.0	19.0	29.7
*Haemophilus influenzae* type b (Hib)	40.0	20.3	54.7
Japanese encephalitis virus	59.9	36.8	85.2
Enterovirus 71 (EV-71)	14.1	31.2	34.4
Rotavirus	14.3	33.5	31.2

* Possible transmission routes include airborne, food-borne, close contact, mosquito-borne, and blood-borne. ** Possible vaccines include meningococcal polysaccharide vaccine (MPSV)/meningococcal polysaccharide conjugate vaccine (MPCV) for meningococcus, 13-valent pneumococcal conjugate vaccine (PCV13)/23-valent pneumococcal polysaccharide vaccine (PPSV23) for *Streptococcus pneumonia*, Hib-containing vaccine for Hib, Japanese encephalitis vaccine (JEV) for Japanese encephalitis virus, EV-71 vaccine for EV-71, and rotavirus vaccine for rotavirus.

**Table 3 vaccines-10-02054-t003:** Scores of knowledge and health beliefs across the demographic groups, history of communicable diseases in children, and vaccine uptake.

Factors	Category	Number of Respondents	Knowledge		Health Beliefs	
			Mean (SD)	*p* Value	Mean (SD)	*p* Value
Family caregivers	Mother	693	27.37 (5.84)	0.5073	56.05 (6.30)	0.1080
	Father	294	26.87 (6.86)		56.99 (6.63)	
	Grandparents	24	27.13 (8.45)		56.05 (6.30)	
Gender	Male	307	26.82 (7.01)	0.2077	56.90 (6.69)	0.0625
	Female	704	27.39 (5.84)		56.08 (6.31)	
Ethnicity	Han Chinese	890	26.82 (7.01)	0.1635	56.51 (6.54)	0.0063
	Minority Chinese	121	27.12 (6.21)		55.02 (5.41)	
Age (year)	≤30	238	26.67 (6.19)	0.0178	56.16 (6.92)	0.3558
	31–35	531	27.03 (6.13)		56.17 (6.01)	
	≥36	242	28.17 (6.37)		56.85 (6.82)	
Children’s age (years)	<3	185	28.56 (6.32)	0.0012	57.43 (6.88)	0.0101
	≥3	826	26.92 (6.16)		56.08 (6.31)	
Number of children	1	513	27.12 (6.21)	0.5981	56.83 (6.29)	0.0111
	≥2	498	27.32 (6.23)		55.81 (6.54)	
Monthly family income (CNY)	<5000	210	26.02 (6.51)	0.0097	54.64 (6.34)	<0.0001
	5000–9999	347	27.63 (6.09)		56.93 (6.14)	
	≥10,000	295	27.71 (6.26)		57.60 (6.56)	
	No disclosure	159	26.98 (5.84)		54.89 (6.25)	
Educational level	Below the college	305	25.64 (6.00)	<0.0001	54.39 (5.96)	<0.0001
	College and above	706	27.90 (6.19)		57.16 (6.45)	
History of communicable diseases	No history	623	26.61 (6.33)	0.0001	55.80 (6.51)	0.0027
	1	245	27.80 (5.99)		56.95 (6.25)	
	≥2	143	28.86 (5.73)		57.57 (6.17)	
Uptake of MPSV/MPCV	Self-paid vaccine	331	27.86 (6.24)	0.0008	57.45 (6.94)	<0.0001
	Free vaccine or unclear	567	27.22 (6.08)		56.14 (6.05)	
	No uptake	113	25.30 (6.47)		54.02 (6.12)	
Uptake of JEV	Self-paid vaccine	208	27.10 (5.92)	0.0621	56.40 (6.31)	<0.0001
	Free vaccine or unclear	673	27.48 (6.22)		56.78 (6.53)	
	No uptake	130	26.08 (6.58)		53.88 (5.55)	
Uptake of four other vaccines	3–4	397	28.45 (6.10)	<0.0001	57.96 (6.90)	<0.0001
	1–2	238	27.94 (5.68)		56.48 (5.77)	
	No uptake	376	25.47 (6.28)		54.50 (5.83)	

SD, standard deviation; CNY, Chinese Yuan; MPSV, meningococcal polysaccharide vaccine; MPCV, meningococcal polysaccharide conjugate vaccine; JEV, Japanese encephalitis vaccine.

**Table 4 vaccines-10-02054-t004:** Factors associated with vaccine uptake across demographic groups, knowledge, and health beliefs.

Factors *	Category	Odds Ratio (OR) and 95% Confidence Interval (CI)					
		MPSV/MPCV	JEV	PCV13/PPSV23	Hib-Containing Vaccine	Rotavirus Vaccine	EV-71 Vaccine
Number of children	1		1.66 (1.30, 2.45)	1.29 (0.99, 1.68)			1.37 (1.05, 1.78)
	≥2		1.00	1.00			1.00
Age (years)	≤30					1.91 (1.31, 2.79)	
	31–35					1.39 (1.01, 1.92)	
	≥36					1.00	
Ethnicity	Han Chinese			2.02 (1.32, 3.10)	2.53 (1.59, 4.00)	1.89 (1.24, 2.89)	1.99 (1.29, 3.07)
	Minority Chinese			1.00	1.00	1.00	1.00
Family income (CNY)	<5000	1.00	1.00	1.00	1.00	1.00	1.00
	5000–9999	1.82 (1.08, 3.05)	2.81 (1.50, 5.26)	1.61 (1.00, 2.61)	1.87 (1.27, 2.76)	1.90 (1.31, 2.76)	1.42 (0.98, 2.06)
	≥10,000	2.96 (1.60, 5.46)	4.07 (2.16, 7.67)	2.83 (1.59, 5.04)	2.98 (1.98, 4.47)	2.44 (1.65, 3.62)	1.85 (1.25, 2.74)
	No disclosure	0.99 (0.57, 1.73)	1.21 (0.70, 2.08)	1.32 (0.85, 2.05)	1.79 (1.14, 2.81)	1.65 (1.06, 2.55)	1.11 (0.72, 1.74)
Educational level	Below the college			1.00	1.00	1.00	1.00
	College and above			0.62 (0.46,0.84)	0.70 (0.51, 0.94)	0.61 (0.45, 0.82)	0.62 (0.46, 0.84)
Knowledge score	Per 1 incremental score	1.04 (1.00, 1.07)		1.05 (1.02, 1.07)	1.05 (1.03, 1.08)	1.02 (1.00, 1.05)	1.06 (1.03, 1.08)
Health belief score	Per 1 incremental score	1.05 (1.01, 1.08)	1.06 (1.03, 1.10)	1.05 (1.03, 1.08)	1.06 (1.03, 1.08)	1.04 (1.02, 1.06)	1.04 (1.02, 1.06)

* Only factors that were statistically significant were presented with OR values and 95% CI in the table. MPSV, meningococcal polysaccharide vaccine; MPCV, meningococcal polysaccharide conjugate vaccine; JEV, Japanese encephalitis vaccine; PCV13, 13-valent pneumococcal conjugate vaccine; PPSV23, 23-valent pneumococcal polysaccharide vaccine; Hib, *Haemophilus influenzae* type b; EV-71, enterovirus 71; CNY, Chinese Yuan.

**Table 5 vaccines-10-02054-t005:** Indices of model fitness for the structural equation model.

	Goodness-of-Fit Index (GFI)	Adjusted Goodness-of-Fit Index (AGFI)	Comparative Fit Index (CFI)	Incremental Fit Index (IFI)	Tucker-Lewis Index (TLI)	Parsimony Normed Fit Index (PNFI)	Root Mean Square Error of Approximation (RMSER)
Fitting value	0.925	0.903	0.933	0.933	0.920	0.770	0.056
Reference value [37]	>0.9	>0.9	>0.8	>0.9	>0.9	>0.5	<0.1

**Table 6 vaccines-10-02054-t006:** Path coefficient in the structural equation model.

Hypothesis	Path Coefficient	*p* Value	Cutoff Ratio
Vaccine uptake ← knowledge	0.354	<0.001	3.355
Vaccine uptake ← perceived susceptibility	0.212	0.010	2.590
Vaccine uptake ← perceived severity	0.039	0.811	0.239
Vaccine uptake ← perceived benefits	−0.804	0.054	−1.956
Vaccine uptake ← self-efficacy	0.584	0.001	3.314
Vaccine uptake ← perceived barriers	0.054	0.589	0.541
Vaccine uptake ← cues to action	0.181	0.238	1.179
Vaccine uptake ← family income	0.159	<0.001	4.899

## Data Availability

The datasets generated during the current study are not publicly available due to privacy but are available from the corresponding author Yihan Lu on a reasonable request.

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
