# Peer review of "Vaccine Uptake to Prevent Meningitis and Encephalitis in Shanghai, China"

_vaccines, 2022, doi:10.3390/vaccines10122054_

Round 1
Reviewer 1 Report
This is an interesting and important analysis of attitudes towards vaccination in China, and generally well presented. While I have a number of areas where the authors could clarify and/or reword, I have serious concerns about the scientific soundness of some of the "correct answers" to the survey questions--and therefore to the interpretability of the results.
Abstract:
Rotavirus only very rarely causes encephalitis—seems strange to include it here; that wouldn’t be the reason anyone recommends the vaccine.
Abstract: what does “self-efficacy” mean? Please re-word.
Association is not necessarily causality: “affected by” is too strong; Lines 294-301 also.
Table 1: Meningococcal polysaccharide vaccines are given at 6 and 9 months of age in China? Usually they are considered ineffective, and possibly inducing hyporesponsiveness, below 2 years of age.
Line 99: Jan-April 2022: the authors should discuss the effect of COVID on vaccine uptake, and whether that influenced their results.
Line 129: How was vaccination history ascertained, and was any attempt made to validate? Was it based only on parenteral assertion? This should be discussed.
Line 143: herpex simplex is an EPI vaccine with uptake of 90% in China? I was not aware such a vaccine exists.
Line 146: what does “common clinical performance” mean? Please re-word.
Line 153: I don’t understand the scoring, perhaps because I’m not sure what the authors consider to be the correct answers. For example, as noted above, to ask parents if rotavirus is a cause of M/E and deduct a point if they don’t know that—frankly, very few infectious diseases specialists, much less physicians, would say that as it is very very rare. Another example: , S. pneumoniae and Hib can each prevent meningitis and pneumonia—does that mean max two points for each? Similarly, the meningococcus can also cause pneumonia as well as meningitis. It would be important for the reviewers to review and validate the “correct” answers.
Line 158: Difficult/impossible to evaluate without knowing what was asked.
Line 193: if inclusion criteria was having children 1-6 years (line 100), why did 14% of the participants not have children 1-6, and were their results excluded?
Figure 1: only a small fraction appears to have gotten DTP-Hib or DTP-IPV-Hib. Is DTP alone on the market? Does that mean coverage with a DTP-containing vaccine is only ~40%? There is a large fraction of “unclear” results—if we assume most of them did get a DTP-containing vaccine, can we assume that the same is true for the “unclear” bars for the other vaccines? What does “unclear” mean?—didn’t answer?
Table 2: How do authors explain discordance between % of people who think a certain pathogen causes M/E and % of people who think a vaccine against that pathogen could prevent M/E? More specifically, except for meningococcus, there appears to be little correlation in those percentages, with sizeable percentages of respondents who don’t think a certain pathogen causes M/E nonetheless believe that a vaccine against the pathogen would prevent M/E. Or am I misreading it?
Figure 3: Why isn’t question C12 about adverse effects included in the “self-efficacy” section with the other adverse effects questions? What does C10 “cost a bit” mean? i.e., in the original Chinese, does that mean it is expensive or not expensive?
Lines 254-259: The percentages described in the text curiously do not match the bar graph percentages in Figure 3 in any case. For example, C2 in the text is given at 23.8%, but in the graph they add up to 21.2; C11 is 28.9 not 31.8, etc..
I question the scientific basis for assuming that for questions C2, C5, and C12 the “correct” answer is to “strongly agree.” There may be nuances in the English translation, but “my child is very susceptible to M/E” seems an overstatement given the rarity of M/E; “M/E vaccines prevent children from infection” is generally not true, though they do prevent disease once infected; “M/E vaccines may have long term effects in children” is certainly possible, even if extraordinarily rare. Again, seeing what the authors considered the “correct” answers is critical.
Table 3: how can there be no difference in grandparents vs parents vis a vis knowledge, but there is a clear difference by age, since grandparents are by definition older than parents?
Lines 284-301, Table 5: These are statistically abstruse concepts that will be well beyond the vast majority of readers, and belong in the supplement, not the main text.
Reviewer 2 Report
Authors of the manuscript titled “Vaccine uptake to prevent meningitis and encephalitis in 2 Shanghai, China” had the aim to analyze the uptake of 6 meningitis and encephalitis vaccines in children and scrutinize the knowledge and health beliefs of their caregivers. In perusing this aim, authors utilized a survey and questionnaires that dealt with demographics, history of communicable diseases in these children, vaccine history, knowledge of the pathogens, transmission routes and clinical outcomes, and health beliefs towards meningitis and encephalitis, meningitis and encephalitis vaccines and vaccination. The methods used and statistical analysis of the data received from respondents is sound and presented in a thoughtful way. The results are displayed logically and thoroughly with ease in supporting the discussion.
Minor comments are provide to make the manuscript more reader friendly and clear.
*Comma need Line 21: Shanghai, China
*Spacing needed Line 197 388(38.38%)
*Order of writing: Line 36-38. Suggest to write the incidence of meningitis first than encephalitis as this order was established in the abstract and should be consistent thorough out the document.
Methods
*Citations are needed for the six vaccines or their manufacturers.
Table 1. Immunization schedules of six vaccines that may prevent meningitis/encephalitis in children.
Order of the vaccine in the table should be labelled and follow the order in the text to allow easier reading and understanding. In addition there are confusing, unaligned * marks in the table. Why is this? Left and justified would clean up the table and make if more easier to read and follow.
Line 99: Authors stated “Inclusion criteria were as follows: 1) 99 family caregivers had children aged 1-6 years; and 2) their children”. Does this mean that family caregivers with one child were omitted? Please clarify?
Line 105: Why did authors state “non-EPI which means lower uptake rate.”? Where is the evidence for this or citation?
Please provide a citation of your procedure you used for the “sample size” or provide reasoning as to the method employed. For example why did the authors use “by using the uptake rate of the four non-surrogate non-EPI vaccines” for sample size determination?
*2.2 Questionnaire: Please provide explicitly the questionnaire in supplementary data or a citation for this questionnaire.
*Line 125. What is meant by “Using a convenient sampling strategy”? Please be explicit.
*2.3 Measurement of outcomes: Do the authors have citations or reasoning to support why demographics and history of communicable in children were measured as ”categorical variables”? And why vaccination history was measured as a ”dichotomous variable”?
And why was this strategy used “In the knowledge section, we utilized multi-choice questions to determine the 140 knowledge?” or is there a citation for this type of questionnaire?
Results
*Where is the raw data from the surveys? Will it be available for others to use? Routinely these datasets are available without the worry of privacy.
*Figure 3 is out of the boards and some information is missing.
Discussion:
*It may be useful for the authors to reflex explicitly to their data in the discussion so the reader can feel confident about the authors comparison statements such as in “In our study, similar factors were determined, such as demographics, knowledge, and health beliefs; of them, family income was significantly associated with uptake of all six M/E vaccines and had largest impact.” And in “Moreover, children who received free and self-paid JEV were more likely to receive free MPSV 332 or self-paid MPCV, respectively, with a moderate consistency. ”, etc. Please put the exact figure or table to support these statements and similar statements in the discussion.
This statement alone needs explicit context Line 338 “We obtained consistent findings in our study.”
Can the authors compare and contrast their results of HBM in this study to others’ studies? As in citations [29,31,47,48].
Reviewer 3 Report
we read the article by Maimaiti et al titled " uptake to prevent meningitis and encephalitis in Shanghai, China" where the authors evaluated the vaccine uptake of pneumococcal vaccine, Hib-containing vaccine, rotavirus vaccine, and EV-71 vaccine as well as the meningococcal vaccine (MPSV/MPCV).
the study itself has excellent statistical analysis , it would have good to elaborate on the methodology and include the steps of performing the "Figure 2. chord diagram" and the "Fig 4. path diagram".
As to the study, the work would benefit from elaborating on the study limitations and proposing future directions on how this work would affect policy and vaccine intake by the population which is absent in the current work.
the language is appropriate and has well flow in reading.
overall, an interesting and very informative study.
Round 2
Reviewer 1 Report
Thank you for addressing this reviewer's concerns